# *Lactiplantibacillus plantarum* HY7718 Improves Intestinal Integrity in a DSS-Induced Ulcerative Colitis Mouse Model by Suppressing Inflammation through Modulation of the Gut Microbiota

**DOI:** 10.3390/ijms25010575

**Published:** 2024-01-01

**Authors:** Hyeon-Ji Kim, Hye-Jin Jeon, Joo-Yun Kim, Jae-Jung Shim, Jae-Hwan Lee

**Affiliations:** R&BD Center, hy Co., Ltd., 22, Giheungdanji-ro 24beon-gil, Giheung-gu, Yongin-si 17086, Republic of Korea; skyatk94@gmail.com (H.-J.K.); 10003012@hy.co.kr (H.-J.J.); jjshim@hy.co.kr (J.-J.S.); jaehwan@hy.co.kr (J.-H.L.)

**Keywords:** *Lactiplantibacillus plantarum*, probiotics, inflammatory bowel disease, colitis, tight junction, anti-inflammation, gut microbiota

## Abstract

Inflammatory bowel disease (IBD), a chronic condition that causes persistent inflammation in the digestive system, is closely associated with the intestinal microbiome. Here, we evaluated the effects of *Lactiplantibacillus plantarum* HY7718 (HY7718) on IBD symptoms in mice with dextran sulfate sodium (DSS)-induced colitis. Oral administration of HY7718 led to significant improvement in the disease activity index score and the histological index, as well as preventing weight loss, in model mice. HY7718 upregulated the expression of intestinal tight junction (TJ)-related genes and downregulated the expression of genes encoding pro-inflammatory cytokines and genes involved in the TLR/MyD88/NF-κB signaling pathway. Additionally, HY7718 reduced the blood levels of pro-inflammatory cytokines, as well as reversing DSS-induced changes to the composition of the intestinal microbiome. HY7718 also increased the percentage of beneficial bacteria (*Lactiplantibacillus* and *Bifidobacterium*), which correlated positively with the expression of intestinal TJ-related genes. Finally, HY7718 decreased the population of pathogens such as *Escherichia*, which correlated with IBD symptoms. The data suggest that HY7718 improves intestinal integrity in colitis model mice by regulating the expression of TJ proteins and inflammatory cytokines, as well as the composition of the intestinal microflora. Thus, *L. plantarum* HY7718 may be suitable as a functional supplement that improves IBD symptoms and gut health.

## 1. Introduction

Inflammatory bowel disease (IBD) affects the health of people worldwide, and its prevalence is increasing [1]. IBD is a chronic disease of unknown cause that results in intestinal inflammation and includes Crohn’s disease (CD) and ulcerative colitis (UC) [2]. CD can cause inflammation throughout the entire digestive tract, from the mouth to the anus, although the inflammation often spreads sporadically to several isolated places. In contrast, inflammation associated with UC is limited to the large intestine, including the colon and rectum [3,4,5]. Patients with IBD exhibit symptoms such as abdominal pain, bloody stools, diarrhea, and weight loss [6]. These physical symptoms can reduce the individual’s quality of life and cause emotional stress; therefore, appropriate treatment is essential [7]. Treatments include steroids and aminosalicylates, but long-term use can cause side effects. Recent studies suggest that probiotics may be a safe treatment, prompting much interest in this area [8,9].

In 2001, the World Health Organization (WHO) defined a probiotic as “Live microorganisms that when administered in adequate amounts confer a health benefit on the host” [10]. Most probiotics comprise lactic acid bacteria (LAB) such as *Lactobacillus*, *Bifidobacterium*, *Lactococcus*, *Enterococcus*, and *Streptococcus* [11]. Among these, *Lactiplantibacillus plantarum* (*L. plantarum*) is found in fermented foods such as kimchi, yogurt, and cheese [12,13]. These genera exert beneficial anti-inflammation, anti-obesity, antioxidant, and anticancer effects, as well as protecting against liver disease [14,15,16,17]. Additionally, *L. plantarum* improves the symptoms of intestinal diseases caused by intestinal barrier disruption and dysbiosis of the gut microbiota [18].

Recent studies show that probiotics improve IBD symptoms in animal models [19,20]. The dextran sulfate sodium (DSS)-induced colitis mouse model is one of the most commonly used in vivo models of human IBD [21]. DSS-induced colitis typically presents as increased colonic epithelial permeability, tight junction (TJ) disruption, and the increased secretion of cytokines such as tumor necrosis factor (TNF)-α, interleukin (IL)-6, IL-1β, and IL-17A [22,23]. Previous studies show that the probiotics *L. fermentum*, *L. gasseri*, and *L. reuteri* not only improve gut barrier function and protect against damage to the colon mucosa but also modulate intestinal immune responses in the DSS-induced colitis model [1,24,25]. Research into the effects of probiotics in mouse models of colitis is ongoing; nevertheless, the mechanism by which probiotics support intestinal barrier function is unclear [26].

In a previous study, we isolated *Lactiplantibacillus plantarum* HY7718, a new lactic acid bacterium with excellent storage stability, from fermented squid, and evaluated its probiotic properties and industrial applicability [27]. Additionally, in vitro experiments using intestinal epithelial cells confirmed that *L. plantarum* HY7718 (HY7718) positively regulates the expression of TJ molecules associated with maintaining intestinal integrity and alleviates oxidative stress and inflammatory responses.

The aim of the present study was to investigate the mechanism by which *L. plantarum* HY7718 supports TJs and suppresses inflammation in mice with DSS-induced colitis. We show histologically that *L. plantarum* HY7718 has positive effects on the colon’s structure and also reveals modulation of intestinal microbiota composition. Finally, we analyzed the correlation between the expression of physiological and genetic biomarkers and the composition of the intestinal flora to investigate the impact of changes in the gut microbiome on IBD symptoms.

## 2. Results

### 2.1. Effects of HY7718 on Physiological Indicators in Mice with DSS-Induced Colitis

To investigate the effects of HY7718 on mice with DSS-induced colitis, we recorded their dietary and water intake and confirmed that all groups consumed similar amounts of food and water containing DSS (Appendix A). In this study, we used sulfasalazine as a positive control; it is known to treat and prevent acute attacks of mild-to-moderate UC [28]. *L. plantarum* ATCC 14917 was used as the lactic acid bacteria control, which is used as a reference strain for probiotic potential tests and is known to be an important strain for the food industry [29,30,31].

Body weight changes, disease activity index (DAI) score, and the length of the colon were measured during the period of DSS administration (Figure 1). The body weight of mice in the control group tended to increase slightly, but that of all mice in the DSS treatment group decreased during the experiment (Figure 1A). Figure 1B shows that the body weight of the DSS group was significantly lower (88.74 ± 4.52%, *p* < 0.001) than that of the control group (103.28 ± 2.88%). Treatment with sulfasalazine (SULF) and *L. plantarum* HY7718 (HY7718) led to significant recovery of body weight, to 93.22 ± 1.80% and 93.23 ± 1.42%, respectively, compared with the DSS group (*p* < 0.05). *L. plantarum* ATCC 14917 (ATCC 14917) led to a slight recovery of body weight to 90.80 ± 2.44%, but the difference compared to the DSS group was not significant.

As shown in Figure 1C, the score for hematochezia in the DSS group was significantly higher than that in the control group (*p* < 0.001). The hematochezia score for the SULF and HY7718 treatment groups was significantly lower than that for the DSS group (*p* < 0.05), whereas the score was not significantly different from the control in the ATCC 14917 group. Therefore, the data suggest that HY7718 significantly downregulates hematochezia compared with ATCC 14917 (*p* < 0.01). The diarrhea score for the DSS group was also significantly higher than that for the control group (Figure 1D, *p* < 0.05); indeed, all treatment groups had a lower diarrhea score than the DSS group (in the order SULF > HY7718 > ATCC 14917), although the reduction was not significant.

In addition, we calculated the DAI score by summing the weight loss, hematochezia, and diarrhea scores during the DSS treatment period (Figure 1E,F). The DAI score for the DSS group was significantly higher (6.36 ± 1.91) than that for the control group (*p* < 0.001). The DAI index for the positive control and probiotic-treated groups (HY7718 and ATCC 14917) was significantly lower (3.42 ± 1.00 (*p* < 0.001), 3.60 ± 1.43 (*p* < 0.001), and 4.70 ± 1.25 (*p* < 0.05), respectively) than that for the DSS-treated group (Figure 1E).

Colon length is a representative marker for evaluating colitis-induced mouse models [32]. Here, we found that colon length decreased to 7.16 ± 0.12 cm in mice treated with DSS (*p* < 0.001) but recovered significantly to 8.10 ± 0.43 cm upon treatment with SULF (*p* < 0.001). HY7718 also led to the recovery of colon length (to 7.83 ± 0.16 cm, *p* < 0.01). In contrast, ATCC 14917 did not result in a significant recovery of colon length (7.52 ± 0.53 cm). Taken together, these results indicate that HY7718 recovers DAI scores and prevents the shortening of the colon at a level similar to that of SULF.

### 2.2. Effects of HY7718 on the Histological Analysis of Colon Tissues

To analyze the effect of HY7718 on colon morphology, we examined sections of the colon tissue samples through the use of hematoxylin and eosin (H&E) staining (Figure 2A). As shown in Figure 2B, the increase in the inflammatory cell infiltration score after treatment with DSS fell significantly after the administration of HY7718 (*p* < 0.05). In contrast, SULF and ATCC 14917 did not reduce inflammatory cell infiltration significantly. Figure 2C shows that crypt damage induced by DSS was alleviated significantly by HY7718 (*p* < 0.001). All treated groups showed significantly lower crypt damage scores than the DSS group (SULF and ATCC 14917, *p* < 0.01; HY7718, *p* < 0.001). Additionally, the histological analysis scores of the SULF, HY7718, and ATCC 14917 groups were significantly lower than those of the DSS group (Figure 2D, SULF and ATCC 14917, *p* < 0.05; HY7718, *p* < 0.01). However, the HY7718 group showed the greatest reduction. Taken together, these results show that HY7718 prevents inflammatory cell filtration, as well as the destruction of the colon mucosa, in the DDS-induced colitis mouse model.

### 2.3. Effects of HY7718 on TJ- and Pro-Inflammatory Cytokine-Related Gene Expression in Colon Tissues

Next, we examined the effect of HY7718 on the expression of genes related to TJ formation and pro-inflammatory cytokines in the DSS-induced colitis model. Intestinal barrier dysfunction results from increased intestinal permeability due to disruption of the TJ barrier [21,33]. As shown in Figure 3A–C, the expression of mRNA encoding *Tjp-1*, *Ocln*, and *Cldn1* (all involved in TJ formation) was significantly lower in the DSS group than in the control group. However, the expression of *Tjp-1*, *Ocln*, and *Cldn1* was restored significantly in the positive control and HY7718 groups. In contrast, ATCC 14917 did not restore the expression of these genes.

The expression *Tnf*, *Il-6*, and *Il-1β* was also analyzed (Figure 3D–F). The expression of mRNA encoding these three cytokines increased in the DSS group but was suppressed in all other treatment groups. In particular, HY7718 led to a significant reduction in the expression of these genes, with results similar to those for SULF (Figure 3D,E). The expression of these genes in the ATCC 14917 group tended to be lower than that in the DSS-induced group, although there was a significant difference only with respect to *Il-1β* expression (Figure 3F).

### 2.4. Effects of HY7718 on the Systemic Secretion of Pro-Inflammatory Cytokines

Next, we measured the levels of TNF-α, IL-6, IL-1β, IL-17A, and IFNγ in serum samples from all groups. The levels of TNF-α, IL-6, IL-1β, and IL-17A were significantly higher in the DSS group than in the control group (Figure 4A–D). The levels of IFNγ were also higher, but the difference was not significant (Figure 4E). The levels of TNF, IL-6, IL-1β, IL-17A, and IFNγ decreased after SULF administration, with reductions in IL-6, IL-17A, and IFNγ being significant (IL-6, *p* < 0.001; IL-17A and IFNγ, *p* < 0.01). The ingestion of ATCC 14917 reduced the levels of all five cytokines to levels similar to those in the positive control SULF group. HY7718 significantly reduced the amounts of TNF (*p* < 0.01), IL-6 (*p* < 0.001), IL-1β (*p* < 0.001), IL-17A (*p* < 0.001), and IFNγ (*p* < 0.05) when compared with DSS. In other words, HY7718 showed the greatest suppressive effect on the secretion of inflammatory cytokines.

### 2.5. Effects of HY7718 on the Expression of TLR/MyD88/NF-κB Signaling Pathway-Related Genes in Colon Tissue

The TLR/MyD88/NF-κB signaling pathway regulates inflammatory signal transduction pathways related to the progression of UC. Activation of TLR4 induces the activation of MyD88 and NF-κB, leading to intestinal inflammation via the secretion of pro-inflammatory cytokines [34]. Also, the expression of TLR4 and TLR2 is upregulated in IBD patients [35]. Here, we found that the expression of *Tlr4*, *Myd88*, *Nfκb1*, and *Tlr2* in the DSS group was significantly higher than that in the control group (Figure 5A–D). HY7718 significantly suppressed the expression of these genes compared with that in the DSS group. ATCC 14917 also reduced the expression of *Tlr4*, *Myd88*, *Nfκb1*, and *Tlr2* mRNA, but only significantly in the case of *Myd88* and *Nfκb1*. Therefore, HY7718 upregulates the expression of TJ-related genes and downregulates the expression of genes related to pro-inflammatory cytokines and the TLR/MyD88/NF-κB signaling pathway more effectively than ATCC 14917.

### 2.6. Effects of HY7718 on the Composition of the Gut Microbiota of Mice with DSS-Induced Colitis

To investigate DSS-induced changes in the microbial population, as well as the effect of HY7718, we analyzed the gut microbiota by targeting 16S rRNA using the Illumina MiSeq platform. To evaluate the alpha diversity of the intestinal flora, we measured the Chao1, Observed, and ACE indices (Figure 6A). The Chao1, Observed, and ACE indices fell in response to DSS but increased in response to HY7718 and ATCC 14917 (Chao1, *p* = 0.0029665; Observed, *p* = 0.0023858; and ACE, *p* = 0.0025744). In contrast, the SULF intake group showed a decrease in the α-diversity indices. Figure 6B shows changes in the β-diversity of the samples (according to the Bray–Curtis index distances between the microbial composition of each group). The principal coordinate analysis (PCoA) showed that the intestinal components of the DSS group were distinct from those of the control group. The SULF group was separated from the other groups. The HY7718 and ATCC 14917 groups were separated, with them lying between the control and DSS groups. As shown in Figure 6C, compared with the control group, the abundance of Proteobacteria was higher in the DSS group (2.13% vs. 13.28%, respectively). In contrast, the abundance of Proteobacteria in the HY7718 and ATCC 14917 groups decreased to 6.32% and 5.58% compared with the DSS group. The abundance of Actinomycetota in the HY7718 group was higher (6.04%) than in the other groups (control, 2.47%; DSS, 3.86%; SULF, 0.21%, and ATCC 14917, 2.41%). Next, we analyzed the A linear discriminant analysis (LDA) score to confirm the differences in the bacterial composition (Figure 6D). There was a significant difference at the microbial family level, with a log LDA score of 4.0. Next, the intestinal microbiota was analyzed at the family level (Figure 6D and Appendix A). Figure 6D shows that compared with the control group, the DSS showed a reduction in the abundance of *Lactobacillaceae* and *Bifidobacteriaceae* and an increase in the abundance of *Enterobacteriaceae* and *Peptostreptococcaceae*. Finally, the gut microbiota was analyzed at the genus level (Figure 6E). Figure 6E shows that compared with the control group, the abundance of *Bifidobacterium* and *Kineothrix* decreased, whereas that of *Escherichia* and *Romuoustia* increased in the DSS group. In contrast, these changes tended to be reversed by SULF and probiotics (HY7718 and ATCC 14917). However, the abundance of *Bifidobacteriaceae* and *Bifidobacterium* increased to a level similar to that in the control group only in the HY7718 group, whereas levels in the SULF and ATCC 14917 groups were similar to those in the DSS group. In addition, the abundance of *Lactiplantibacillus* increased in the HY7718 and ATCC 14917 groups. In particular, mice receiving HY7718 had a higher abundance of *Lactiplantibacillus* than those receiving ATCC 14917. Taken together, these data suggest that HY7718 supports the gut diversity and intestinal bacterial components destroyed by DSS.

### 2.7. Correlation Heatmap between Relative Abundance and Biochemical Indicators

Finally, we performed Pearson’s correlation analysis of microbial abundance and biochemical indicators. As shown in Figure 7, the DAI score, histological analysis score (inflammation and crypt damage), and inflammatory cytokine levels correlated positively with pathogenic (phylum, Proteobacteria; family, *Enterococcaceae* and *Peptostreptococcaceae*; genus, *Eshcerichia,* and *Romboutsia*) bacteria but negatively with beneficial bacteria (family, *Bifidobacteriaceae* and *Lactobacillaceae*; genus, *Bifidobacterium*, *Lactiplantibacillus* and *Kineothrix*). In contrast, colon length and expression of TJ-related genes *Tjp-1*, *Ocln*, and *Cldn1* correlated positively with host beneficial bacteria but negatively with pathogenic bacteria. These results show that HY7718 modulates bacterial composition to improve the intestinal barrier, inflammation, and other parameters.

## 3. Discussion

IBD is a representative intestinal disease with increasing prevalence worldwide [1]. Recent studies show that probiotics such as *Lactobacillus* species are a safe treatment for IBD; indeed, data suggest that probiotics improve colitis in the DSS-induced mouse model and IBD patients by regulating the intestinal barrier, immune responses, and gut microbiota [36,37]. Recently, we demonstrated that the probiotic strain HY7718 suppresses inflammatory responses and maintains TJs between intestinal epithelial cells [27]. In the present study, we show that HY7718 improves intestinal integrity, reduces intestinal mucosal damage, and regulates the gut microbiota in a DSS-induced colitis mouse model. We also show a correlation between intestinal microflora composition and other indicators such as physiological indicators, colonic histological index, and TJs-related gene expression and pro-inflammatory markers.

The mice were fed 2% DSS water for 5 days to induce acute UC, followed by a water recovery period of 2 days. SULF (which is used to treat acute attacks of mild-to-moderate UC) was used as a positive control [28]. DSS drinking induced various symptoms, including weight loss, rectal bleeding, and loose stools, in the mice during the experiment, as well as shortening the length of the large intestine. Mice receiving DSS alone showed a gradual increase in the DAI index (calculated by summing weight loss, hematochezia, and diarrhea scores) during the DSS drinking period and showed no improvement even during the water recovery period. In contrast, the DSS-induced increase in the DAI index was significantly reduced during the water recovery period in mice receiving SULF or probiotics. In addition, SULF and probiotics (HY7718 and ATCC 14917) restored colon length. In particular, the effect of HY7718 on the DAI score and colon length was similar to that of the positive control (SULF) and superior to that of type strain ATCC 14917. Colitis induced by DSS results in intestinal inflammation and damage to the crypts in the colonic mucosa [38]. H&E staining of colon tissues demonstrated that the administration of SULF or probiotics to mice with DSS-induced colitis reduced intestinal inflammatory cell infiltration and mucosal crypt damage. In particular, the lowest levels of inflammatory cell infiltration were observed in the group receiving HY7718.

Reduced expression of TJ-related genes in mice with DSS-induced colitis confirmed that destruction of the intestinal barrier occurs actively in this model. TJs play an important role in intestinal barrier function. The critical components of TJs are tight junction protein-1 (*Tjp-1*), occludin (*Ocln*), and claudin-1 (*Cldn1*) [39]. Here, we confirmed that HY7718 restored the expression of mRNA encoding TJ-related genes in TNF-treated intestinal epithelial cells, as reported previously [27]. We also confirmed that HY7718 restored the expression of TJ-related genes, which was suppressed by DSS. These results suggest that HY7718 has the potential to enhance barrier function in UC and CD by restoring disrupted TJs.

In general, the balanced expression of pro-inflammatory cytokines in the colonic mucosa is essential for maintaining intestinal homeostasis. However, IBD (including CD and UC) is associated with cytokines disturbance due to the overproduction of pro-inflammatory cytokines [40]. Increased inflammatory responses triggered by DSS were confirmed by analyzing the expression of genes in the colon. In addition, we investigated the levels of TNF, IL-6, IL-1β, IL-17A, and IFNγ in serum. According to previous studies, pro-inflammatory cytokines promote the dysfunction of epithelial TJs and increase intestinal permeability [41,42]. The results reported herein show that HY7718 suppresses inflammatory cytokine production, suggesting that this probiotic improves disrupted intestinal permeability in cases of colitis.

Toll-like receptors (TLRs) activate signaling pathways to induce the expression of inflammatory and immune-related genes [43]. Previous studies show that cryptepithelial cells isolated from the mucosa of IBD patients show higher expression of TLR4 than cells from healthy individuals. In addition, TLR4 and TLR2 are overexpressed in the intestinal epithelium of patients with IBD, resulting in excessive inflammatory responses [35,44,45]. The TLR/MyD88 pathway mediates the activation of NF-κB. Increased expression of NF-κB damages the colonic mucosa directly by inducing the production of pro-inflammatory cytokines such as TNF, IL-6, and IL-1β [46]. Therefore, we investigated inflammatory responses induced by HY7718 in the colitis mouse model by examining the TLR/MyD88/NF-κB signaling pathway. We confirmed that HY7718 reduced the expression of genes encoding *Tlr4*, *MyD88*, *Nfκb1*, and *Tlr2*, which was increased by DSS. Our data, including histological analysis, gene expression analysis, and cytokines secretion studies, suggest that HY7718 suppresses inflammatory responses by regulating the TLR/MyD88/NF-κB signaling pathway in IBD.

Finally, we analyzed the diversity and gut microbiota in each group of mice and then investigated the correlation between the relative abundance of different populations and other indicators. According to various studies, the diversity and abundance of the gut microbiota in IBD patients are different from those in healthy people [47,48,49]. Generally, α-diversity in IBD is lower than that in healthy individuals [50]. Here, we found that DSS reduced α-diversity (i.e., Chao1, Observed, and ACE), whereas probiotics (HY7718 and ATCC 14917) recovered diversity to levels similar to that in the control. Analysis of β-diversity revealed some distance between the control and DSS groups, but the HY7718 group was closer to the control than the DSS group. Proteobacteria, which include various pathogenic genera such as *Escherichia*, *Enterobacter*, *Vibrio*, *Salmonella*, and *Yersinia*, disrupt the microbial composition of the intestine in colitis model mice and IBD patients [51]. *Escherichia* (belonging to the Proteobacteria phylum and *Enterobacteriaceae* family) are typical pathogens that promote intestinal inflammation and mucosal bacterial dysbiosis in mice with colitis [51,52]. An increase in the *Peptostreptococcaceae* (including *Romboutsia*) population has been reported in IBD, including UC [53]. Here, we show that HY7718 reduces the abundance of Proteobacteria at the phylum level, that of *Enterobacteriaceae*, *Peptostreptococcaceae* at the family level, and that of *Enterobacter*, *Escherichia* and *Romboutsia* at the genus level. In addition, these pathogenic bacteria correlated positively with levels of inflammatory cytokines (TNF, IL-6, IL-1β, IL-17A, and IFNγ) and genes involved in inflammatory signal transduction pathways (*Tlr4*, *Myd88*, *Nfκb1*, and *Tlr2*). *Bifidobacterium* is a representative bacterium (belonging to the *Bifidobacteriaceae* family and Actinomycetota phylum) that is beneficial to the host because it maintains the intestinal mucosal barrier [54]. Administration of HY7718 increased the abundance Actinomycetota at the phylum level, *Bifidobacteriaceae* at the family level, and *Bifidobacterium* at the genus level. These results suggest that Bifidobacterium correlates positively with the expression of colonic TJ proteins. Thus, HY7718 alleviates intestinal barrier dysfunction by regulating the composition of the gut microbiota. Furthermore, the abundance of the *Lactiplantibacillus* genus was higher in the probiotic intake group, with HY7718 having a greater effect than ATCC 14917. This suggests that HY7718 colonizes the intestine to a greater extent than ATCC 14917. These data are consistent with our previous study showing that HY7718 adheres well to epithelial cells [27]. *Kineothrix* is a recently discovered bacterium that produces butyrate. IBD reduces the production of short-chain fatty acids such as butyrate; butyrate-producing bacteria are associated with increased intestinal homeostasis and reduced inflammation [55,56]. Here, we found that *Kineothrix* abundance correlated negatively with the expression of inflammation-related markers. HY7718 treatment has the potential to improve intestinal inflammation by restoring the abundance of *Kineothrix*, which is reduced by DSS. Taken together, the results of the gut microbiota analyses indicate that HY7718 colonizes the intestine, regulates microbial diversity, prevents colonization by pathogenic bacteria, and increases the proportion of beneficial strains in mice with colitis.

## 4. Materials and Methods

### 4.1. Bacterial Culture and Sample Preparation

*Lactiplantibacillus plantarum* HY7718 (HY7718) was isolated from Korean fermented squid and maintained at −80 °C as frozen stocks in MRS broth (BD Difco, Sparks, MD, USA) containing 20% (*v*/*v*) glycerol. HY7718 was cultivated in MRS agar broth at 37 °C for 48 h under anaerobic conditions and then centrifuged at 4000× *g* for 20 min. The cell pellet was washed twice and resuspended in sterile phosphate-buffered saline for in vitro assays. For the animal experiments, fresh cultured HY7718 was freeze-dried and supplied as feed. *Lactiplantibacillus plantarum* ATCC 14917 (ATCC 14917) was used as a LAB control to compare in vitro and in vivo effects, and ATCC 14917 was prepared the same way as HY7718.

### 4.2. Animal Experiments

Six-week-old male C57BL/6 mice were purchased from Raonbio (Yongin-si, Gyeonggi-do, Republic of Korea). The animals were maintained at a room temperature of 22 ± 1 °C, with 55 ± 10% humidity and a 12 h light/dark cycle. After 1 week of adaptation, the mice were randomly assigned to five groups (10 mice per group): non-treatment (control); DSS 2% (DSS); DSS 2% + sulfasalazine (SULF); DSS 2% + *L. plantarum* HY7718 (HY7718); or DSS 2% + *L. plantarum* ATCC 14917 (ATCC 14917).

The mice were orally administered 100 μL of sulfasalazine (100 mg/kg/day) and 100 μL of the probiotic (10^9^ CFU/kg/day) over 14 days; the control and DSS groups were orally administered an equal volume of saline over the same period. After starting the administration of probiotics for 7 days, the mice were given 2% DSS (MP Biomedicals, Santa Ana, CA, USA; molecular weight = 36,000–50,000 Da) in their drinking water to induce colitis; the exception was the control group. Body weight and food and water intake were measured daily during the study period. Blood, colon, and cecum samples were harvested at the end of the study for further analysis. All animal experiments were approved by the Ethics Review Committee of R&BD Center, hy Co., Ltd., Republic of Korea (AEC-2023-0004-Y). A flow chart of the animal studies is shown in Figure 8.

### 4.3. DAI Scoring

The DAI scores were calculated as described in a previous study [57]. The degree of body weight loss was scored from 0 to 6 (0, <0%; 1, 0–5%; 2, 6–10%; 3, 11–15%; 4, 16–20%; 5, 21–25%; and 6, 26–30%). The degree of diarrhea was scored from 0 to 3 (0, normal stools; 1, soft stools; 2, loose stools; and 3, watery stools). The degree of hematochezia was scored from 0 to 3 (0, negative fecal occult blood; 1, small amounts of blood-streaked feces; 2, conspicuous blood-wrapped feces; and 3, visible rectal bleeding).

### 4.4. Histological Analysis

The length of the collected colon was measured. In addition, seven samples of colon per group were selected for histological analysis. Colon tissues were fixed in 10% formalin solution (Sigma Aldrich, St. Louis, MO, USA), embedded in paraffin, sectioned, and stained with H&E. Images were obtained under a Zeiss Axiovert 200M microscope (Carl Zeiss AG, Thornwood, NY, USA). The histological score, which includes the severity of inflammation, the extent of inflammation, crypt damage, and percent involvement, was evaluated as described previously [38]. The score for inflammatory cell infiltration was calculated as the sum of severity and the extent of inflammation. The severity of inflammation was scored from 0 to 3 (0, none; 1, mild; 2, moderate; 3, severe). The extent of inflammation was scored from 0 to 3 (0, none; 1, mucosa; 2, mucosa and submucosa; 3, transmural). The degree of crypt damage was scored from 0 to 4 (0, none; 1, basal 1/3 damaged; 2, basal 2/3 damaged; 3, crypts with lost-surface epithelium present; 4, crypts and surface epithelium lost). Then, the histological analysis score was calculated by multiplying the score for the percent involvement to analyze the colon tissue of each specimen. Percent involvement (%) was scored from 0 to 4 (0, 0; 1, 1–25; 2, 26–50; 3, 51–75; 4, 76–100). Histological colitis analysis of H&E staining and scoring were performed by Doo Yeol Biotech (Seoul, Republic of Korea).
Histological analysis score = each specimen score × percentage involvement

### 4.5. Extraction of Total RNA and Gene Expression Analysis

Total RNA was extracted from mouse colon tissues using the easy-spin Total RNA Extraction Kit (iNtRON Biotechnology, Seoul, Republic of Korea), and cDNA synthesis was conducted at 37 °C for 60 min using the Omniscript Reverse Transcription Kit (Qiagen, Hilden, Germany). The synthesized cDNA was amplified by a QuantStudio 6-Flex Real-time PCR System (Applied Biosystems, Foster City, CA, USA). Real-time PCR was conducted using the Taqman Gene Expression Master Mix and TaqMan Gene expression assays (Applied Biosystems, Foster City, CA, USA). The target genes and the TaqMan probes are listed in Table 1. *Gapdh* was used for normalization.

### 4.6. Measurement of Inflammatory Cytokines

The BD OptEIA™ Mouse IL-6 ELISA Set (BD Biosciences, San Diego, CA, USA; BD 555240) was used to measure the levels of inflammatory cytokines IL-6, IL-1β, TNFα, IFNγ and IL-17A in the mouse serum. Absorbance at 450 nm was measured on a BioTek^®^ Synergy HT Microplate reader (Santa Clara, CA, USA). The serum levels of IL-1β, TNFα, IFNγ, and IL-17A were measured at LABISKOMA (Seoul, Republic of Korea) via multiplex analysis.

### 4.7. DNA Extraction and Quantification

Total genomic DNA (gDNA) was extracted from mouse cecum tissue using a QIAamp DNA Stool Mini Kit (Qiagen, Hilden, Germany). The extracted DNA was quantified using a Quant-iT PicoGreen dsDNA Assay Kit (Invitrogen, Waltham, MA, USA).

### 4.8. Library Construction and Sequencing

Sequencing libraries were prepared according to the Illumina 16S Metagenomic Sequencing Library protocols to amplify the V3 and V4 regions. The input gDNA (5 ng) was PCR-amplified in 5× reaction buffer, 1 mM of dNTP mix, 500 nM each of the universal forward/reverse (F/R) PCR primer, and Hercules II fusion DNA polymerase (Agilent Technologies, Santa Clara, CA, USA). The cycling conditions for the first round of the PCR were as follows: 3 min at 95 °C for heat activation, followed by 25 cycles of 30 s at 95 °C, 30 s at 55 °C, and 30 s at 72 °C, followed by a 5 min final extension at 72 °C. The following universal primer pair with Illumina adaptor overhang sequences was used for the first amplification:V3-F: 5′-TCGTCGGCAGCGTCAGATGTGTATAAGAGACAGCCTACGGGNGGCWGCAG-3′, V4-R: 5′-GTCTCGTGGGCTCGGAGATGTGTATAAGAGACAGGACTACHVGGGTATCTAATCC-3′

The amplicon from the first round of the PCR was purified using MPure beads (Agencourt Bioscience, Beverly, MA, USA). Following purification, 2 μL of the PCR product was amplified using the Nextera XT Index Kit (Illumina, San Diego, CA, USA) to construct the final library containing the index. The cycling conditions for the second PCR were the same as for the first round, except that 10 cycles were performed. The PCR product was purified with AMPure beads and quantified using a KAPA Library Quantification Kit (KAPA Biosystems, Wilmington, MA, USA) and the Agilent D1000 ScreenTape (Agilent Technologies, Waldbronn, Germany). Paired-end (2 × 300 bp) sequencing was performed by Macrogen (Seoul, Republic of Korea) using the MiSeq^TM^ platform (Illumina, San Diego, CA, USA).

### 4.9. Bioinformatics Analysis

After Illumina MiSeq sequencing was complete, the Illumina MiSeq raw data for each sample were classified using the index sequence, and paired-end FASTQ files were created. Using the Curadapt program (v3.2), a preprocessing step was performed to remove the adapter sequences and the F/R primer sequences from the target gene region; the forward (Read1) and reverse sequences (Read2) were then trimmed to 200–250 bp each. The DADA2 package (v 1.18.0) of the R program (v 4.0.3) was used for error correction during the amplicon sequencing process. Sequences with an expected error of ≥2 paired-end reads were excluded. The data that completed the preprocessing process were used to establish an error model for each batch to remove noise from each sample. After assembling the paired-end sequences with corrected sequencing errors into a single sequence, chimeric sequences were removed using the consensus method in DADA2, and amplicon sequence variants (ASVs) were formed. Among the generated ASVs, those shorter than 350 bp were excluded by the R program (v 4.0.3). Bioinformatics analysis was performed using Microbiomanalyst (https://www.microbiomeanalyst.ca/, accessed on 13 November 2023). Alpha diversity was measured using the Chao1, Observed, and ACE indices. Beta diversity was calculated using the Bray-Curtis distance between samples and visualized in three-dimensional plots. Taxonomy abundance was analyzed according to the SILVA database. A linear discriminant analysis (LDA) was performed to identify significantly different phylotypes among the experimental groups. The correlation between gut microbiota and biochemical indicators was calculated via the Pearson coefficient and visualized using the pheatmap package. A correlation heatmap was constructed using R software (v 4.0.3). All datasets have been deposited in the NCBI Sequence Read Archive (SRA) with the accession code PRJNA1043834.

### 4.10. Statistical Analysis

All data, except the microbiome analysis, are expressed as the mean ± standard deviation (SD). Differences between groups were analyzed by one-way ANOVA followed by Tukey’s post hoc test. The statistical analysis was performed with GraphPad Prism 6.0 software (GraphPad Software, San Diego, CA, USA). *p* < 0.05 was considered significant. For the bioinformatics analysis, differences between groups were analyzed by a *t*-test/ANOVA using MicrobiomAnalyst.

## 5. Conclusions

The data presented herein suggest that *Lactiplantibacillus plantarum* HY7718 has potential as a functional supplement for improving colitis. HY7718 alleviates physiological indicators such as the DAI index, colon length, and histological scores (including inflammatory cell infiltration and crypt damage). In addition, HY7718 improves intestinal barrier function by regulating the expression of tight junction protein-related genes. Furthermore, HY7718 exerts anti-inflammatory effects by suppressing the secretion of pro-inflammatory cytokines and expression of genes involved in the TLR/MyD88/NF-κB signaling pathway. Finally, HY7718 modulates intestinal microbial composition and diversity in mice with DSS-induced colitis. Thus, we propose the use of HY7718 by the food industry as a functional supplement that alleviates colitis and regulates the gut microbiota.

## Figures and Tables

**Figure 1 ijms-25-00575-f001:**
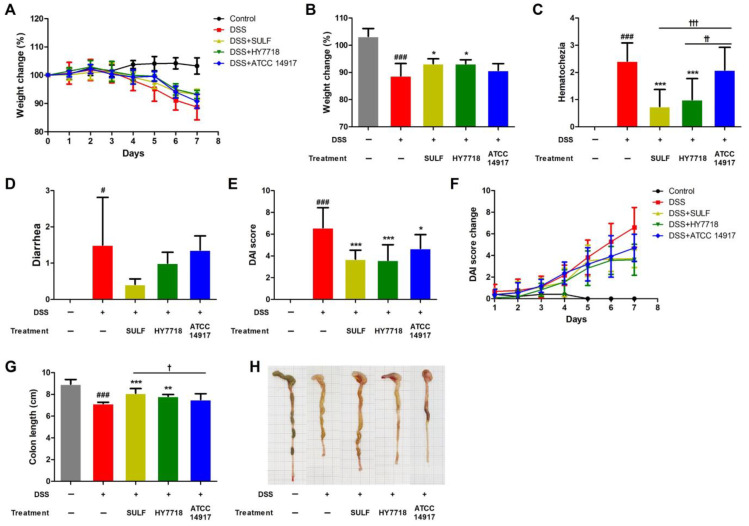
Effect of HY7718 on physiological indicators in mice with DSS-induced colitis. (**A**) Change in body weight, (**B**) weight loss (%), (**C**) hematochezia, (**D**) diarrhea, (**E**) DAI score, (**F**) DAI score, (**G**) colon length, and (**H**) colon tissue morphology. The results are presented as the mean ± SD. Significant differences are indicated as ^#^ *p* < 0.05 and ^###^ *p* < 0.001 compared to the control group; * *p* < 0.05, ** *p* < 0.01, and *** *p* < 0.001 compared to the DSS group; and ^†^ *p* < 0.05, ^††^ *p* < 0.01 and ^†††^ *p* < 0.001 among the treatment groups. DSS, dextran sulfate sodium; SULF, sulfasalazine; HY7718, *Lactiplantibacillus plantarum* HY7718; ATCC 14917, *Lactiplantibacillus plantarum* ATCC 14917; DAI, disease activity index.

**Figure 2 ijms-25-00575-f002:**
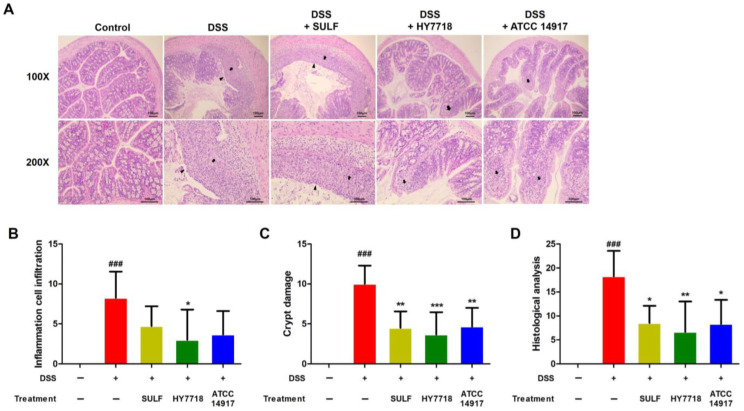
Effect of HY7718 on histology, as shown by the analysis of H&E-stained tissue sections from mice with DSS-induced colitis. (**A**) Histological changes (H&E stained, 100× (top) and 200× (bottom) for magnification). Arrows and arrowheads indicate crypt damage, inflammation cell infiltration and erosion of the epithelium, (**B**) inflammatory cell infiltration, (**C**) crypt damage, and (**D**) histological analysis score. The results are presented as the mean ± SD. Significant differences are indicated as ^###^ *p* < 0.001 compared to the control group; * *p* < 0.05, ** *p* < 0.01, and *** *p* < 0.001 compared to the DSS group. DSS, dextran sulfate sodium; SULF, sulfasalazine; HY7718, *Lactiplantibacillus plantarum* HY7718; ATCC 14917, *Lactiplantibacillus plantarum* ATCC 14917.

**Figure 3 ijms-25-00575-f003:**
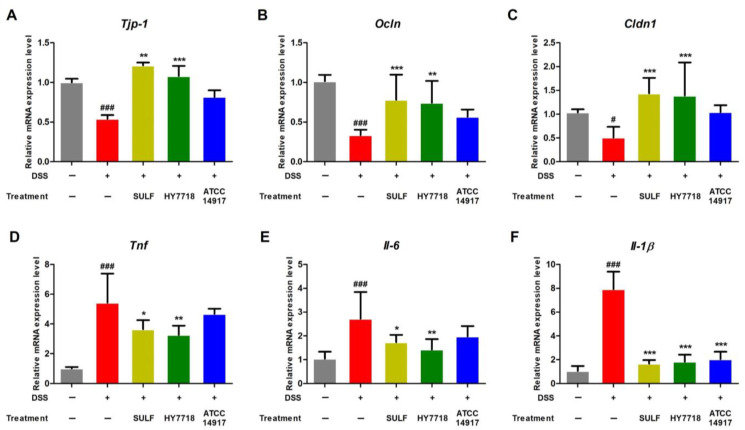
Effect of HY7718 on the expression of genes in the colon of mice with DSS-induced colitis. Expression of genes encoding tight junction proteins (**A**) Tjp-1, (**B**) Ocln, and (**C**) Cldn1, and the expression of genes encoding (**D**) Tnf, (**E**) Il-6, and (**F**) Il-1β. The results are presented as the mean ± SD. Significant differences are indicated as ^#^ *p* < 0.05, and ^###^ *p* < 0.001 compared to the control group and * *p* < 0.05, ** *p* < 0.01, and *** *p* < 0.001 compared to the DSS group. DSS, dextran sulfate sodium; SULF, sulfasalazine; HY7718, *Lactiplantibacillus plantarum* HY7718; ATCC 14917, *Lactiplantibacillus plantarum* ATCC 14917; Tjp-1, tight junction protein-1; Ocln, occludin; Cldn1, claudin-1; Tnf, tumor necrosis factor-α; Il-6, interleukin-6; Il-1β, interleukin-1 beta.

**Figure 4 ijms-25-00575-f004:**
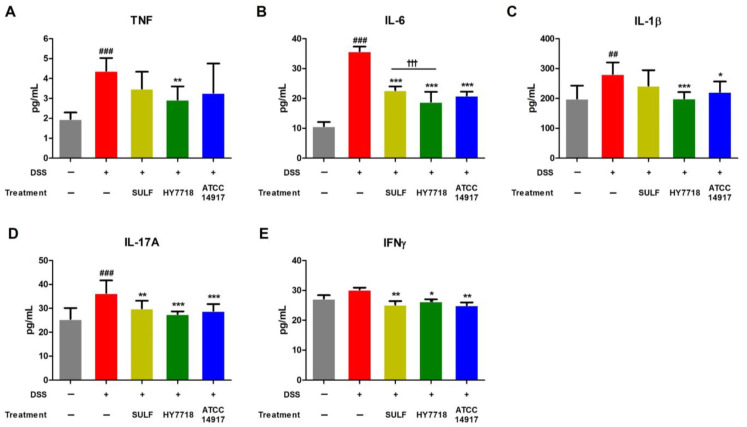
Effect of HY7718 on the secretion of pro-inflammatory cytokines in DSS-induced colitis mice. Concentration of (**A**) TNF, (**B**) IL-6, (**C**) IL-1β, (**D**) IL-17A, and (**E**) IFNγ. The results are presented as the mean ± SD. Significant differences are indicated as ^##^ *p* < 0.01, and ^###^ *p* < 0.001 compared to the control group, * *p* < 0.05, ** *p* < 0.01, and *** *p* < 0.001 compared to the DSS group, and ^†††^ *p* < 0.001 among the treatment groups. DSS, dextran sulfate sodium; SULF, sulfasalazine; HY7718, *Lactiplantibacillus plantarum* HY7718; ATCC 14917, *Lactiplantibacillus plantarum* ATCC 14917; TNF, tumor necrosis factor-α; IL-6, interleukin-6; IL-1β, interleukin-1 beta; IL-17A, interleukin-17A; IFNγ, interferon gamma.

**Figure 5 ijms-25-00575-f005:**
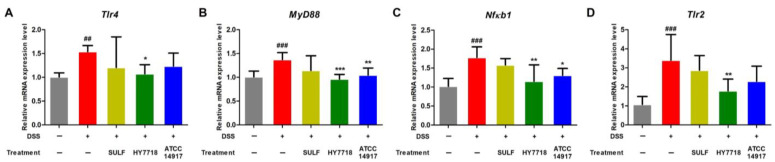
Effect of HY7718 on the expression of genes in the colon of DSS-induced colitis mice. Expression of genes related to the TLR/MyD88/NF-κB signaling pathway: (**A**) Tlr4, (**B**) MyD88, (**C**) Nfκb1, and (**D**) Tlr2. The results are presented as the mean ± SD. Significant differences are indicated as ^##^ *p* < 0.01, and ^###^ *p* < 0.001 compared to the control group and * *p* < 0.05, ** *p* < 0.01, and *** *p* < 0.001 compared to the DSS group. DSS, dextran sulfate sodium; SULF, sulfasalazine; HY7718, *Lactiplantibacillus plantarum* HY7718; ATCC 14917, *Lactiplantibacillus plantarum* ATCC 14917; Tlr4, toll-like receptor 4; MyD88, myeloid differentiation primary response 88; Nfκb1, nuclear factor kappa subunit; Tlr2, toll-like receptor 2.

**Figure 6 ijms-25-00575-f006:**
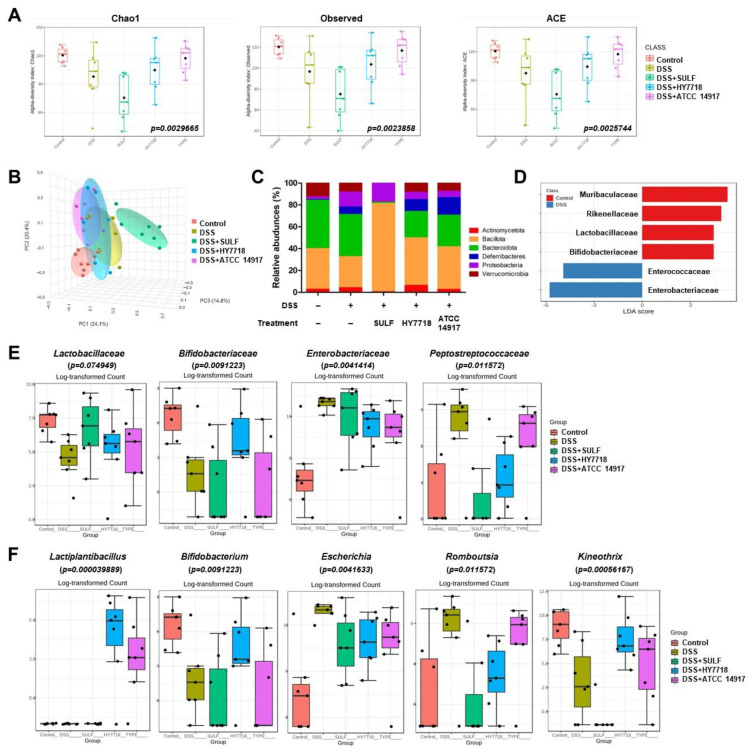
Composition of the intestinal microbiota in DSS-induced colitis mice. (**A**) α-diversity, (**B**) 3D PCoA plot of bacterial β-diversity based on the Bray-Curtis index distance, (**C**) relative abundance at the phylum level, (**D**) LDA score between the control and DSS groups at the family level, (**E**) relative abundance at the family level, and (**F**) relative abundance at the genus level. DSS, dextran sulfate sodium; SULF, sulfasalazine; HY7718, *Lactiplantibacillus plantarum* HY7718; ATCC 14917, *Lactiplantibacillus plantarum* ATCC 14917.

**Figure 7 ijms-25-00575-f007:**
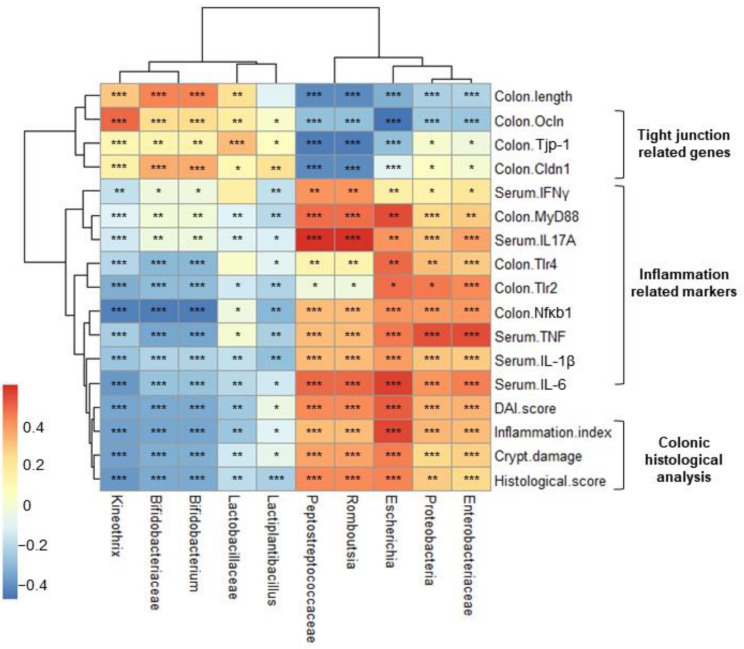
Analysis of the correlation between the gut microbiota and biochemical indicators. The size and color of the circles indicate the corresponding index levels and values, with blue indicating a negative value and red indicating a positive value. The *p*-value was calculated using Pearson’s correlation analysis; * *p* < 0.05, ** *p* < 0.01, and *** *p* < 0.001. DAI, disease activity index; Tjp-1, tight junction protein-1; Ocln, occludin; Cldn1, claudin-1; TNF, tumor necrosis factor-α; IL-6, interleukin-6; IL-1β, interleukin-1 beta; IL-17A, interleukin-17A; IFNγ; interferon gamma; Tlr4, toll-like receptor 4; MyD88, myeloid differentiation primary response 88; Nfκb1, nuclear factor kappa subunit; Tlr2, toll-like receptor 2.

**Figure 8 ijms-25-00575-f008:**
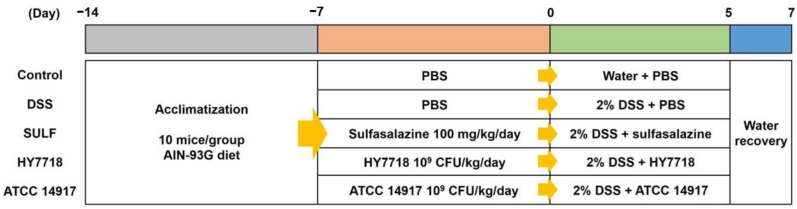
Flow chart showing the animal studies.

**Table 1 ijms-25-00575-t001:** TaqMan probes used for the analysis of mRNA expression, along with their catalog numbers.

Gene	Gene Name	Catalog Number
*Gapdh*	Glyceraldehyde-3-phosphate dehydrogenase	Mm99999915_g1
*Tjp-1*	Tight junction protein 1	Mm01320638_m1
*Ocln*	Occludin	Mm00500910_m1
*Cldn1*	Claudin-1	Mm01342184_m1
*Tnf*	Tumor necrosis factor α	Mm00443258_m1
*Il-6*	Interleukin 6	Mm00446190_m1
*Il-1* *β*	Interleukin 1 beta	Mm00434228_m1
*Tlr4*	Toll-like receptor 4	Mm00445273_m1
*MyD88*	Myeloid differentiation primary response 88	Mm00440338_m1
*Nf* *κ* *b1*	Nuclear factor kappa B subunit1	Mm00476361_m1
*Tlr2*	Toll-like receptor 2	Mm00442346_m1

## Data Availability

The data presented in this study are available in the article and Appendix A.

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
