# Peer review of "Lactiplantibacillus plantarum HY7718 Improves Intestinal Integrity in a DSS-Induced Ulcerative Colitis Mouse Model by Suppressing Inflammation through Modulation of the Gut Microbiota"

_ijms, 2024, doi:10.3390/ijms25010575_

Round 1

Reviewer 1 Report

Comments and Suggestions for Authors

Reviewer 2 Report

Comments and Suggestions for Authors

The current study investigates the impact of a novel probiotic in treating colitis in mice. The study is thoroughly conducted and of a very high standard. The findings are significant and demonstrate the potential for the probiotic to treat colitis / IBD by reducing pro-inflammatory responses and positively modulating the gut microbiome. I recommend publication after the following minor corrections:

1. The authors should report the animal study based on ARRIVE guidelines, in particularly the Essential 10. It would be of interest to know whether the animals were randomized and whether the handling / research team were blinded to various treatment groups.

2. Statistical singificance is reported against control groups in most figures, but it would be good to demonstrate statistical significance across each treatment group. E.g. does ATCC149177 suppress the expression of pro-inflammatory cytokines to a greater degree than SULF?
